# Oxidative Stress Contributes to Bacterial Airborne Loss of Viability

Henry P. Oswin,[a] Allen E. Haddrell,[a] Cordelia Hughes,[a] Mara Otero-Fernandez,[a] Richard J. Thomas,[b] Jonathan P. Reid[a]

aSchool of Chemistry, Cantock's Close, University of Bristol, Bristol, United Kingdom
bDefence Science Technology Laboratory (DSTL), Porton Down, Salisbury, United Kingdom

**ABSTRACT** While the airborne decay of bacterial viability has been observed for decades, an understanding of the mechanisms driving the decay has remained elusive. The airborne transport of bacteria is often a key step in their life cycle and as such, characterizing the mechanisms driving the airborne decay of bacteria is an essential step toward a more complete understanding of microbial ecology. Using the Controlled Electrodynamic Levitation and Extraction of Bioaerosols onto a Substrate (CELEBS), it was possible to systematically evaluate the impact of different physicochemical and environmental parameters on the survival of *Escherichia coli* in airborne droplets of Luria Bertani broth. Rather than osmotic stress driving the viability loss, as was initially considered, oxidative stress was found to play a key role. As the droplets evaporate and equilibrate with the surrounding environment, the surface-to-volume ratio increases, which in turn increased the formation of reactive oxygen species in the droplet. These reactive oxygen species appear to play a key role in driving the airborne loss of viability of *E. coli*.

**IMPORTANCE** The airborne transport of bacteria has a wide range of impacts, from disease transmission to cloud formation. By understanding the factors that influence the airborne stability of bacteria, we can better understand these processes. However, while we have known for several decades that airborne bacteria undergo a gradual loss of viability, we have not previously identified the mechanisms driving this process. In this work, we discovered that oxygen surrounding an airborne droplet facilitates the formation of reactive oxygen species within the droplet, which then gradually damage and kill bacteria within the droplet. This discovery indicates that adaptations to help bacteria deal with oxidative stress may also aid their airborne survival and be essential adaptations for bacterial airborne pathogens. Understanding the adaptations bacteria need to survive in airborne droplets could eventually lead to the development of novel antimicrobials designed to inhibit their airborne survival, helping to prevent the transmission of disease.

**KEYWORDS** *Escherichia coli*, aerobiology, airborne microorganisms, bioaerosols, oxidative damage

There are a variety of mechanisms by which bacteria may be transported through the environment, and understanding these mechanisms is key to understanding bacterial ecology. While many forms of motility through liquids and across solid surfaces have been characterized down to the molecular level (1–5), there remain many unknowns surrounding the transport of bacteria through the air. Airborne bacteria take on a variety of forms, ranging from respiratory pathogens and commensals contained in exhaled droplets of saliva and sputum (6–8), to soil bacteria attached to particles suspended high in the upper atmosphere (9–11). The conditions these bacteria face will be unlike anything they encounter in bulk liquid solutions and as such will be likely to induce a unique physiological response in the bacteria. This physiological

Address correspondence to Henry P. Oswin, ho1451@bristol.ac.uk, or Jonathan P. Reid, J.P.Reid@bristol.ac.uk.

The authors declare no conflict of interest.

response remains largely uncharacterized and poorly understood. Limitations to our understanding of the airborne physiology of bacteria has contributed to uncertainty surrounding important issues such as airborne disease transmission (12), environmental contamination (13), and the contribution of bacteria to cloud formation (14–16).

Evidence of the harmful nature of the airborne microenvironment can be seen in many studies of bacterial airborne survival. Studies using rotating drums (17–19), the Tandem Aged Respiratory Droplet Investigation System (TARDIS) (20), and Controlled Electrodynamic Levitation and Extraction of Bioaerosols onto a Substrate (CELEBS) (21, 22), have reported a time dependent loss of culturability in bacteria exposed to airborne conditions for a variety of different bacterial species under a variety of different conditions. However, the exact mechanisms driving this loss of viability remain uncertain.

It should be possible to identify likely causes of airborne loss of bacterial viability through examination of the unique physicochemical properties of aerosols and airborne droplets. For aqueous airborne droplets, the high surface area-to-volume ratio results in a rapid equilibration of the water activity of the droplet to the surrounding relative humidity (RH) (23–26). In many cases, such as in exhaled aerosols, this equilibration will result in a concurrent reduction in the size of the particle, with water quickly partitioning from the droplet until the water activity has reduced such that it is equilibrated to a drier surrounding RH. For example, a droplet of saliva with a diameter of 35 $\mu$m exposed to 50% RH (typical indoor humidity) would be expected to decrease in size to an equilibrated diameter of ~10 $\mu$m, in approximately 3 s (23), while the solute concentration and surface to volume ratio of the droplet would both increase. The lack of heterogeneous nucleation sites in an airborne droplet means that salt concentrations can become supersaturated (27, 28), exposing bacteria within the droplet to concentrations of solute that they would not encounter in any other environment. Although, it is possible that the presence of microorganisms within a droplet could provide heterogeneous nucleation sites for salt crystallization (29, 30). At a sufficiently low RH, salts can crystallize via homogenous nucleation (31), a process termed efflorescence, embedding any suspended bacteria in the airborne salt crystal (21). Bacteria in equilibrated airborne droplets could be expected to experience osmotic stress because of the low water activity, harmful bacteria-bacteria interactions, starvation, increased exposure to various forms of radiation, and increased exposure to gaseous reactants. Any of these things are potentially responsible for airborne loss of bacterial viability.

The aim of this study is to identify the primary mechanistic explanation for a loss of bacterial viability in aerosols. *Escherichia coli* in droplets of LB broth is the system considered, with both the bacteria and solution well characterized, allowing our study to focus on aerosol specific phenomena. The CELEBS instrument (21, 22, 32, 33) is used for the measurements of airborne stability. CELEBS allows for a high degree of control over the conditions experienced by an airborne microorganism, making it an ideal tool for systematic evaluation of the impact of a range of parameters on the airborne survival of bacteria. Through the systematic elimination of parameters that do not impact the airborne survival of *E. coli* in LB broth and the identification of those parameters that do have an impact, it is possible to establish a probable mechanistic cause for the loss of bacterial viability observed in airborne droplets.

## RESULTS

**Characterization of the response of *E. coli* to the airborne environment.** Two strains of *E. coli* were used in the measurements in this study: MRE162 and K12. MRE162 has been used for airborne survival studies for many years (21, 22, 34–36) and its airborne survival is particularly well characterized, making it ideal for a mechanistic study such as this. As such, MRE162 is the strain used in most of the measurements performed. K12 is also a well-characterized strain commonly used in laboratory study of *E. coli* and is used in this study to explore the impact of deleting various genes on the airborne survival of *E. coli*. The airborne viability of both strains is compared over

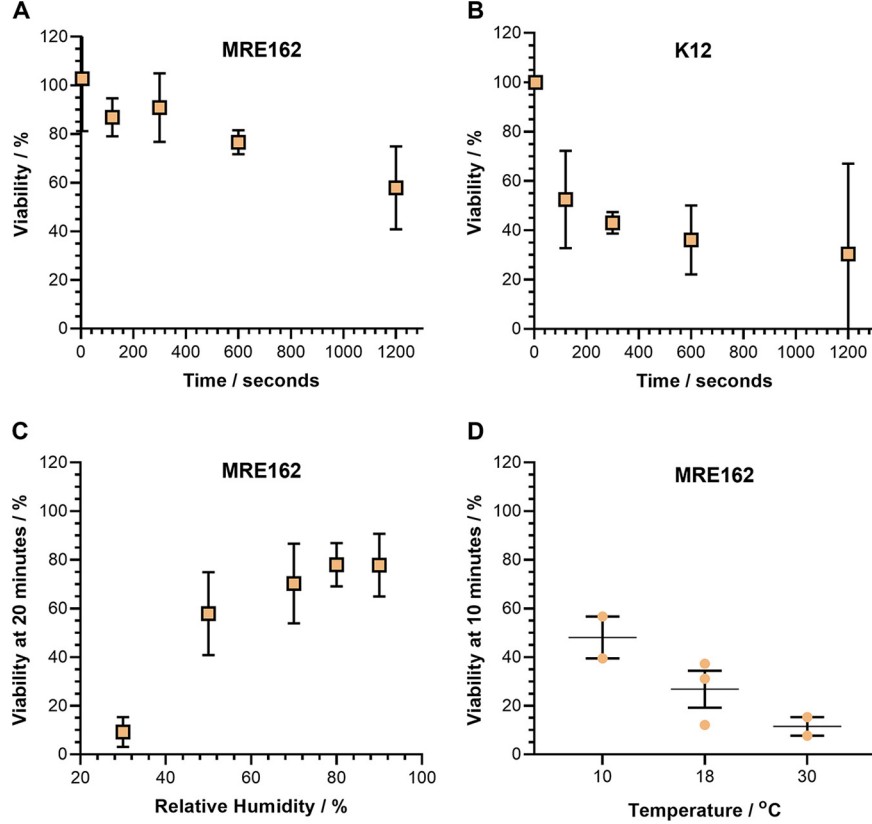

**FIG 1** General characterization of the airborne survival of *E. coli*. CELEBS measurements of the airborne survival of *E. coli* MRE162 and K12 in droplets of LB broth. For A to C datapoints show the mean, error bars show the standard deviation. (A) The airborne survival of *E. coli* MRE162 over 20 min at 50% ± 1% RH, room temperature (18 to 21°C). For 120, and 600 s $n$ = 3, for 300 s $n$ = 9, for 1,200 s $n$ = 13. (B) The airborne survival of *E. coli* K12 over 20 min at 50% ± 1% RH, room temperature (18 to 21°C). For 120 s $n$ = 18, for 300, and 1,200 s $n$ = 3, for 600 s $n$ = 6. (C) The relationship between relative humidity and the survival of *E. coli* MRE162 after 20 min of levitation at room temperature (18 to 21°C). For 30% and 80% RH $n$ = 3, for 50% RH $n$ = 13, for 70% RH $n$ = 6, for 90% RH $n$ = 11. (D) The relationship between temperature and the survival of *E. coli* MRE162 after 10 min of levitation at 30% ± 1% RH. Each point shows the result of an individual measurement. The middle line indicates the mean with the error bars showing the standard deviation. For 10°C and 30°C $n$ = 2, for 18°C $n$ = 3.

the course of 20 min of levitation in LB broth droplets at 50% RH in Fig. 1A and B. Both strains show a loss of culturability during 20 min of levitation at 50% RH, but K12 is significantly less stable ($P$ = 0.009 at 120 s, $P$ = 0.0002 at 300 s, and $P$ = 0.002 at 600 s), falling to a mean viability of 30% ± 37% in 20 min (Fig. 1B), while MRE162 decreased to a mean of 57% ± 17% (Fig. 1A). It should be noted that the 50% RH survival curve for MRE162 presented here differs from a CELEBS generated survival curve for the same conditions published in 2020. This is the result of several improvements to the CELEBS technique that have taken place since those measurements were made (Fig. S1).

A relationship with RH is observable in the airborne stability of MRE162 in LB broth droplets, with a mean viability of 9% ± 6% after 20 min at 30% RH and 78% ± 12% at 90% RH (Fig. 1C). This RH influence is in qualitative agreement with previous studies of bacterial airborne viability that show a positive correlation between RH and the viability of the bacteria (19, 37–39). Although, an opposite trend is occasionally reported in older studies (40). In addition, a clear impact of temperature on the viability of airborne MRE162 was observable. At 18°C (room temperature) in LB broth droplets at 30% RH, the mean viability after 10 min was 27% ± 13%. When the temperature was increased to 30°C, this viability decreased to 11% ± 5%, and when the temperature was decreased to 10°C the viability increased to 48% ± 12% (Fig. 1D). A similar relationship between temperature and the airborne survival of *E. coli* has been previously observed (41).

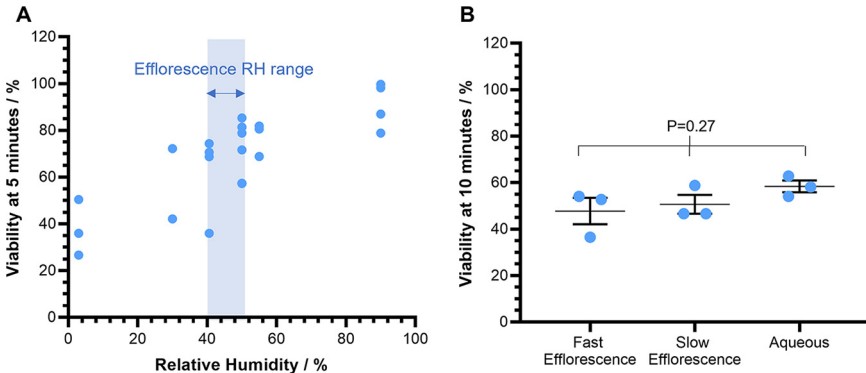

**FIG 2** The impact of efflorescence on the airborne survival of *E. coli*. CELEBS measurements of the airborne survival of *E. coli* MRE162 in droplets of PBS at room temperature (18 to 21°C). (A) The relationship between RH and the survival of *E. coli* MRE162 after 5 min of levitation. The RH range within which the efflorescence threshold for PBS will be crossed is indicated by the blue shaded area. Each point shows the result of an individual measurement. For 3% and 55% RH *n* = 3, for 30% RH *n* = 2, for 40%, and 90% RH *n* = 4, for 50% RH *n* = 6. (B) The impact of the physical state of the equilibrated particle on the survival of *E. coli* MRE162 after 10 min of levitation at room temperature. Fast efflorescence RH was set to 21.7% and all particles effloresced within 5 s of generation; slow efflorescence RH was set to 31.7% and all particles effloresced between 7 and 120 s after generation; the aqueous particles the RH was set to 46.7%. Each point shows the result of an individual measurement. The middle line indicates the mean with the error bars showing the standard deviation. *n* = 3 for all measurements.

The observations reported in Fig. 1 provide the starting point for an investigation of the mechanism that drives the loss of viability of *E. coli* in airborne LB broth droplets. The relationship between temperature and the loss of viability indicates that the mechanism of viability loss is, at least in part, thermodynamically driven. The mechanism is evidently specific to the aerosol form of LB broth, as it is well established that *E. coli* do not lose viability in bulk solutions of LB broth at these temperatures (42). It is therefore unsurprising that the loss of viability is greater at lower relative humidity, as the factors that differentiate airborne droplets from bulk solutions are exacerbated as the RH is lowered. For example, increased loss of water from the airborne droplet will drive increased dehydration of the bacteria, increased exposure of bacteria to solutes within the droplet, and increased interaction of the droplet with gaseous reactants at the air-liquid interface, any of which could contribute the observed loss of viability. At 30% RH and below, the salts within droplets of LB broth become sufficiently concentrated to effloresce via homogenous nucleation (21, 27), which was initially suspected to drive the particularly rapid loss of viability at low RH.

**Airborne droplet efflorescence does not appear to impact the viability of *E. coli*.** Previous CELEBS measurements with SARS-CoV-2 have demonstrated that the relationship between RH and viral airborne stability is explained, in part, by an immediate 50% to 60% loss of infectious virus upon the efflorescence of the droplet at low RH (33). However, droplet efflorescence does not appear to drive a similar loss of viability in CELEBS measurements of *E. coli* (21). While the loss of SARS-CoV-2 infectivity occurred almost immediately as the droplet crystallized, the loss of *E. coli* MRE162 viability was not readily observable until after 1 min of levitation at 30% RH, despite LB broth efflorescing within seconds at 30% RH (21, 27). To further explore the impact of phase on bacterial survival, *E. coli* MRE162 was levitated in PBS droplets at RHs both above and below the efflorescence threshold. PBS was used in place of LB broth as the lack of organic material in PBS means that it can be expected to effloresce in a more complete and reproducible manner than LB broth (21, 27), and, therefore, its efflorescence was expected to be more likely to drive a loss of viability.

The remaining viability from 5-min levitations of *E. coli* in PBS droplets at gradually decreasing relative humidity are reported in Fig. 2A. The typical relationship between RH and airborne survival can be seen in PBS with survival being highest at 90% RH and lowest survival at 3% RH. There is some variation in the precise RH at which a solution will be seen to effloresce (43), and so a range in RH at which PBS may reach the

threshold water activity for efflorescence to occur is indicated in Fig. 2A, with efflorescence always taking place below this range but never above it. While the relationship between RH and survival is observable, no stepwise drop in survival was observed as the RH crossed the efflorescence threshold, providing no evidence of phase change driving an increased loss of viability.

To further investigate the consequences of efflorescence for viability, a similar experiment was carried out with 10-minute levitations. During this measurement, droplet phase was inferred from changes in the light scattering intensity and positional stability of the droplet observed with the camera viewing the trapped droplets from above the ring electrodes in the CELEBS (Fig. 2B). At the highest RH (47%), the droplets remained aqueous throughout the measurement, and the average viability remained at 58% $\pm$ 4%. At the intermediate RH (32%) all droplets were observed to effloresce between 7 s and 2 min after droplet generation, and the average viability after 10 min was 51% $\pm$ 7%. At the lowest RH (22%) all droplets were observed to effloresce within 5 s of generation and the average viability was 48% $\pm$ 10% after 10 min. Again, while the RH trend is observable, no significant difference was observed between the decreases in viability that take place in the effloresced droplets compared to the aqueous droplets. Rather than the efflorescence driving the viability loss, it may instead be postulated that, in the increasingly desiccated conditions as RH is lowered, an efflux of water from the bacteria could be responsible for the loss of viability.

**Dehydration is unlikely to contribute to the airborne loss of viability of *E. coli*.** The requirement for moisture to allow bacterial growth is well understood, with *E. coli* needing a water activity of 0.95 or more (44), equivalent to droplets equilibrated to 95% RH. However, growth inhibition does not equate to a permanent loss of viability in bacteria. It is thought that reduced cellular hydration of bacteria results in damage to proteins, lipids, and nucleic acid through a variety of mechanisms (45, 46). However, past studies measuring desiccation induced losses of *E. coli* culturability tend to do so over too long a timescale to be applicable to the rapid airborne loss of viability reported here (47), making it difficult to determine whether the low water activity of airborne droplets could drive the airborne loss of viability of *E. coli*.

To measure the potential impact of low water activity ($a_w$) on *E. coli* MRE162, the bacteria were resuspended in two bulk aqueous sodium chloride solutions, one with an $a_w$ of 0.95, which is not expected to negatively impact the survival of *E. coli*, and the other with salt dissolved to the bulk saturation limit, giving it an approximate $a_w$ of 0.76 (Fig. 3A). A small reduction in the mean CFU count was observable in both solutions over 3 h, both falling to approximately 80% of the initial count, but there was no significant difference in culturability of *E. coli* between the two suspensions. It is clear that at humidities of 76% or greater, other factors must account for a significant proportion of the loss of viability, with the airborne viability of *E. coli* MRE162 falling below 80% in only 20 min, even at 90% RH.

The disaccharide trehalose serves as an osmoprotectant for *E. coli* and its production rate is increased in response to low water activity conditions (48–50). It can be hypothesized that deletion of the OtsA gene (encoding trehalose-6-phosphate synthase) in *E. coli*, shown to significantly lower intracellular concentrations of trehalose (51–53), would diminish airborne survival if osmotic stress is a major contributor to airborne loss of viability in *E. coli*. However, for the time points studied, the K12 ΔOtsA appeared to have higher viability than K12 WT (Fig. 3B) at 50% RH, remaining at 78% $\pm$ 17% viability after 2 min while K12 WT fell to 52% $\pm$ 19% under the same conditions. It is possible that the improvement to survival could be the result of increased persister cell formation in OtsA deletion mutants (54). The exact mechanism by which trehalose shields *E. coli* from osmotic stress is not fully characterized, although it is thought to act as a molecular chaperone, forming hydrogen bonds with biological macromolecules and stabilizing them (55). The absence of a detrimental impact of OtsA deletion on airborne survival in this experiment indicates that it is unlikely that airborne loss of viability in *E. coli* is caused by the same molecular damage against

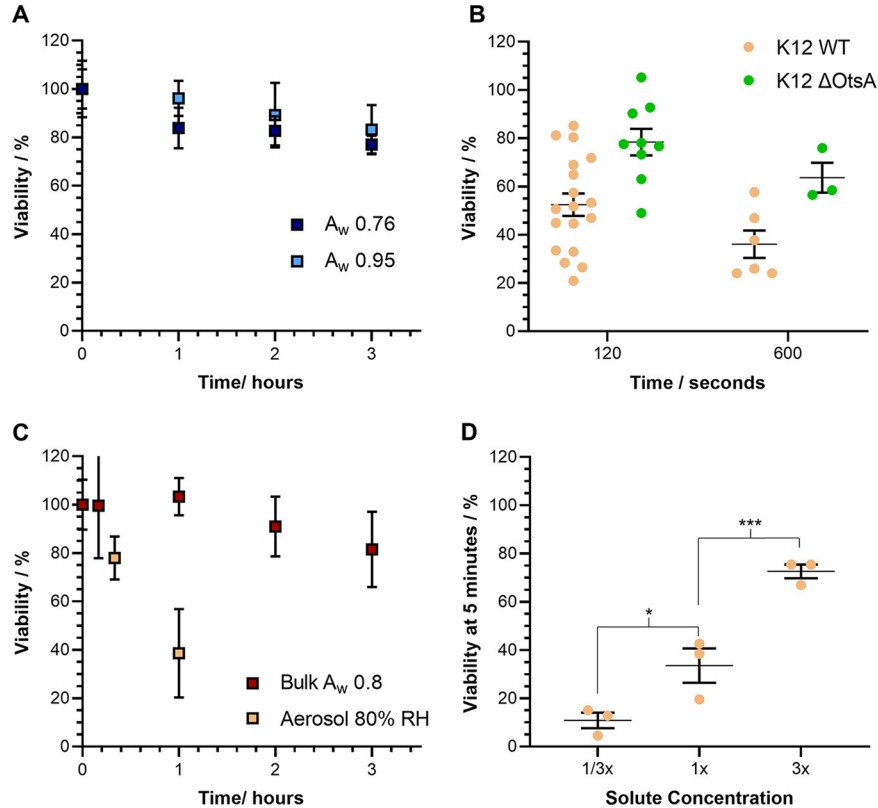

**FIG 3** Exploring the relationship between RH and *E. coli* airborne survival. CELEBS and bulk measurements of the survival of *E. coli* MRE162 and K12 at room temperature (18 to 21°C). A and C datapoints show the mean, error bars show the standard deviation. B and D datapoints show the results of individual measurements, the middle line shows the mean, the error bars show the standard deviation. (A) Survival of *E. coli* MRE162 over 3 h in bulk solutions of NaCl. The light blue points show the survival in a more dilute solution with a water activity of 0.95 and the dark blue points show the survival in a more concentrated solution with a water activity of 0.76. *n* = 5 for all measurements. (B) The airborne survival of *E. coli* K12 (beige points) and an OtsA deficient mutant of *E. coli* K12 (green points) after 2 and 10 min of levitation at 50% ± 1% RH, in LB broth droplets. (C) Comparison of the survival of *E. coli* MRE162 after 1 h of levitation in LB broth droplets at 80% RH (lighter beige points labeled Aerosol 80% RH) to the survival over 3 h in a bulk solution of concentrated LB broth matching the concentrations of all components in an airborne droplet equilibrated to 80% ± 1% RH (orange points labeled bulk Aw 0.8). For all measurements in the bulk solution *n* = 6, for the CELEBS measurement *n* = 3. (D) The airborne survival of *E. coli* MRE162 after 5 min at 30% ± 1% RH, in LB broth droplets with either a third or three times the initial solute concentration compared to the survival in LB broth with a normal initial solute concentration; *, $P < 0.05$; ***, $P < 0.001$.

which trehalose should protect the cell. Taken alongside the difficulty in inducing a loss of viability through bulk exposure to lowered $a_w$, it appears unlikely that dehydration, driven by the low water activity of airborne droplets, plays a major role in the airborne loss of viability of *E. coli*.

**Loss of airborne viability correlated with surface area-to-volume ratio.** It is only possible in bulk LB broth solutions to replicate the concentrations of components in an airborne droplet equilibrated to high RH (equivalent to $a_w$) of 80% and above. Solutes spontaneously form solute crystals below this water activity and the solute concentrations in bulk solution are sustained at their solubility limit. At 80% RH, the volume of a droplet of LB broth decreases by a factor of approximately 15 (21) from the typical starting volume generated at the high water activities of stock solutions used in CELEBS ($a_w$ 0.995). Similar behavior occurs in any release of aerosol from an initially dilute solution at high water activity (e.g., from an aqueous slurry or on exhalation of aerosols from the respiratory tract). The volume change concentrates all solutes within the droplet by the same factor and the concentrations of the key components

of LB broth are approximately 150 g/L NaCl, 150 g/L tryptone, and 75 g/L yeast extract at $a_w \sim 0.8$.

*E. coli* MRE162 was suspended in a concentrated broth at $a_w \sim 0.8$, also with a bacterial concentration 15 times that typically prepared for levitations, with the viability monitored by CFU count. This measurement can then be compared to a CELEBS levitation of *E. coli* MRE162 in LB broth at 80% RH, as the concentrations of all components within the droplets were the same and should any of those components drive the airborne loss of viability, the same loss of viability would be observable in this bulk solution (Fig. 3C). However, the viability of the bacteria remained at 100% after the first hour, and only fell to 81% ± 16% after 3 h in the bulk suspension. For comparison, the viability fell to an average of 39% ± 18% after only 1 h in droplets levitated at 80% RH. This measurement provides considerable insight into the mechanism driving the airborne loss of viability of *E. coli* in LB broth, as the only differences between the bulk solution and levitated droplets in this experiment were the dynamic changes undergone by the airborne droplet during the initial equilibration of the droplet and the final ratio of the surface area-to-volume of the volume confining the bacteria. The 5-mL bulk solution in a test-tube with a radius of 2 cm has a surface area-to-volume ratio of approximately $2.5 \times 10^{-4}$ (1:4,000). An airborne droplet with a radius of 25 $\mu$m has an initial surface area-to-volume ratio of $1.2 \times 10^{-1}$ (1:8), which increases to $3 \times 10^{-1}$ (1:3) when equilibrated to 80% RH. The increased loss of viability in airborne droplets may be related to the higher surface area-to-volume ratio. Increased loss of viability with increasing surface area-to-volume ratio could also explain the role of RH, as a lower RH will result in more water loss from the droplet, and therefore, a lower equilibrated size and higher surface area-to-volume ratio.

Disentangling the impacts of droplet size change (i.e., surface area-to-volume ratio) and solute concentration is possible by altering the initial concentration of a solution before aerosolizing it. Five-min levitations were carried out at 30% RH in LB broth droplets of either a normal concentration (25 g/L) diluted to a third of the normal concentration (8.3 g/L) or at 3 times the normal concentration (75 g/L). These three formulations levitated at the same RH all equilibrate to the same solute concentrations, but to different final droplet sizes, necessarily losing differing amounts of water to reach the same equilibrated concentration. As a minor consideration, the different starting concentrations will also lead to differences in the evaporation kinetics although these differences are small and occur over a time period during which no loss of viability is observed (see Fig. S2 for simulations of the evaporation kinetics of these different starting solution droplets using previously collected physicochemical data [21]).

The difference in equilibration size has a profound effect on the viability of the bacteria, with a clear positive correlation between equilibrated size and viability observed (Fig. 3D). In the droplets with the more dilute-starting formulation, the droplet radius at equilibration can be estimated to be 5.8 $\mu$m (surface area-to-volume ratio of $5.2 \times 10^{-1}$) and the viability fell to 11% ± 6% after 5 min. In the normal formulation, the radius at equilibration is 8.3 $\mu$m (surface area-to-volume ratio of $3.6 \times 10^{-1}$) and the viability fell to 34% ± 12% after 5 min. With the more concentrated LB broth, the radius at equilibration is 12 $\mu$m (surface area-to-volume ratio of $2.5 \times 10^{-1}$) and the average viability remained at 73% ± 5% after 5 min. This same trend has also previously been observed using the CELEBS technique at 50% RH (21). All the evidence collected at this point indicates that the surface area-to-volume ratio of the airborne droplet is a key parameter that correlates with the rate of loss of viability for *E. coli*.

**E. coli airborne loss of viability is reduced in a hypoxic environment.** The apparent link between the surface area-to-volume ratio of the equilibrated droplet and the rate of viability loss suggests that exposure to a gas phase component may drive the loss of viability. To determine if the presence of oxygen has an impact on the loss of viability, CELEBS measurements were repeated with the airflow replaced by a flow of pure nitrogen (Fig. 4A). In all cases, replacing the gas flow with nitrogen increased the average airborne viability of *E. coli* MRE162, but only at 30% RH was a significant

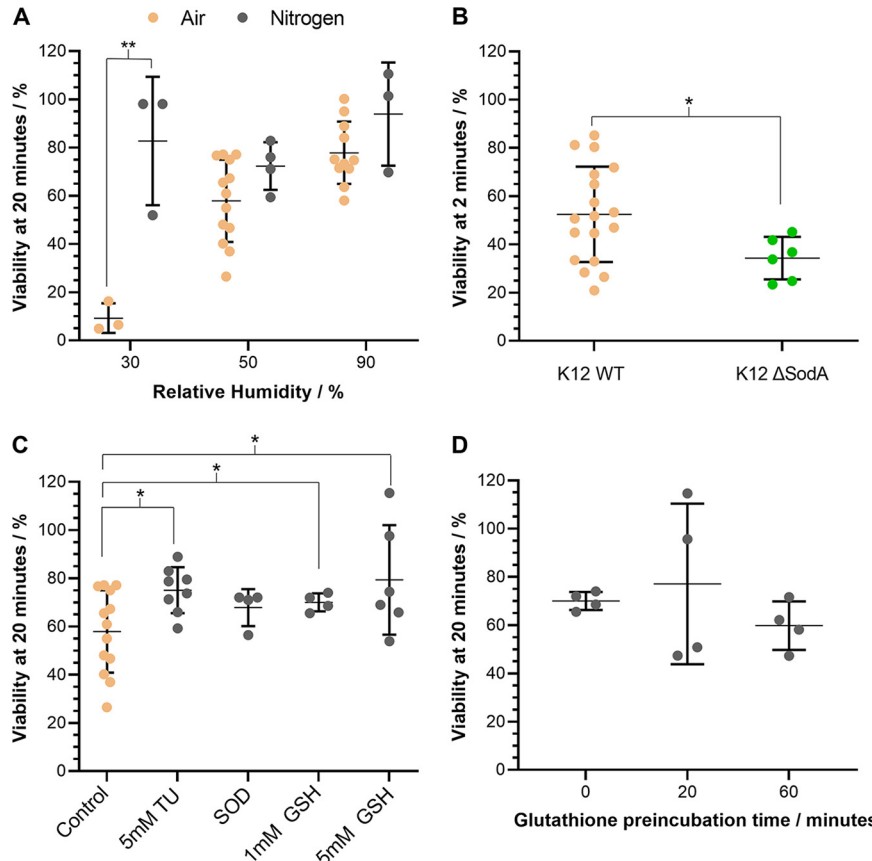

**FIG 4** The impact of oxidative stress on the airborne survival of *E. coli*. CELEBS measurements of the airborne survival of *E. coli* MRE162 and K12 in droplets of LB broth at room temperature (18 to 21°C). Datapoints show the results of individual measurements, the middle line shows the mean, error bars show the standard deviation. (A) Comparison of the airborne survival of *E. coli* MRE162 after 20 min of levitation in air (beige points), to levitations for 20 min in nitrogen (gray points), at 30% ± 1%, 50% ± 1%, and 90% ± 1% RH. **, $P < 0.01$. (B) The airborne survival of *E. coli* K12 (beige points) and a SodA deficient mutant of *E. coli* K12 (green points) after 2 min of levitation at 50% ± 1% RH. *, $P < 0.05$. (C) The effect of the presence of 5 mM thiourea (5 mM TU), 1 unit/mL superoxide dismutase (SOD), 1 mM glutathione (1 mM GSH), and 5 mM glutathione (5 mM GSH) on the airborne survival of *E. coli* MRE162 after 20 min of levitation at 50% ± 1% RH. Measurements are compared to 20 min of levitation at 50% ± 1% RH in LB broth without antioxidants (Control). *, $P < 0.05$. (D) The effect of preincubation at 37°C of a suspension of *E. coli* MRE162 in LB broth with 1 mM glutathione, on subsequent airborne survival measurements of 20 min at 50% ± 1% RH using the suspension.

difference observable ($P < 0.01$). While a clear impact of RH is observable in the presence of air, this trend is not observed in the nitrogen gas flow. Thus, the clearest impact of changing the gas phase from air to nitrogen is observed at the lowest RH, with the average viability after 20 min in 30% RH air being 9% ± 6% and the average viability in 30% RH nitrogen being 83% ± 27%. These results are in agreement with the findings of early rotating drum studies, in which a decreased airborne loss of viability was observed in the absence of oxygen for both *E. coli* and *Serratia marcescens* (17–19, 56), as well as a more recent study of the effects of spray drying on *Lactococcus lactis* (57).

The apparently intertwined impacts of droplet size and oxygen presence on the loss of viability further narrows down the potential causes of the loss of viability. It has been demonstrated in a bulk phase study that aerobic conditions increase *E. coli* loss of viability driven by high salt concentrations and low pH (58). It could be suggested that the high surface area to volume ratio results in a higher concentration of oxygen in smaller droplets. However, the diffusion rate of oxygen at room temperature within a droplet of the sizes studied here is such that the concentration of oxygen will be

maintained at saturation concentration (determined by Henry's law solubility) throughout the levitation and homogeneously throughout the droplet, regardless of droplet size and bacterial metabolism within the droplet (59). Thus, the apparent impact of droplet size on survival cannot be explained by differences in the concentration of dissolved oxygen (Fig. 3D). Furthermore, the loss of viability observed in the bulk solutions (Fig. 3A and C) was very slow. While the oxygen concentration in those bulk solutions will potentially have been below saturation, they will not have been anaerobic. If the loss of viability was a combination of the high concentration of solutes, bacteria, and the presence of oxygen, a more significant loss of viability would likely have been observed in the bulk solution of Fig. 3C. It seems instead that the mechanism of airborne viability loss requires a combination of the presence of oxygen and the high surface area-to-volume ratio of airborne droplets. Previous studies have demonstrated unique chemistry at air liquid interfaces (60–62), and it is possible that oxygen is able to form products harmful to the bacteria through such surface chemistry.

**Reactive oxygen species formation drives airborne loss of viability in *E. coli*.** A K12 $\Delta$SodA mutant was found to have significantly reduced survival after a 2-min levitation at 50% RH compared to K12 WT, with a reduction in viability from 52% $\pm$ 20% to 34% $\pm$ 9% ($P < 0.05$) (Fig. 4B). SodA encodes one of three superoxide dismutases expressed by *E. coli* and its deletion renders the bacteria less able to process superoxide radicals, as demonstrated by an increased susceptibility to paraquat (63, 64). However, while an increased loss of airborne viability when SodA is deleted could indicate a role of superoxide radicals in the airborne loss of viability, it is also possible that reducing the capacity of the bacteria to process reactive oxygen species (ROS) merely introduced oxidative stress as an additional mechanism of viability loss. It was, however, noteworthy that of all the isogenic mutants of K12 studied, it was only the $\Delta$SodA mutant that demonstrated reduced airborne viability (Fig. S3).

To further explore the role of ROS in the airborne loss of viability of *E. coli*, several antioxidants were added to the droplets prior to levitation. The antioxidants tested were: thiourea, a scavenger of superoxide and hydroxyl radicals (65); purified superoxide dismutase from bovine liver; and glutathione, a naturally occurring thiol capable of quenching reactive oxygen species both directly and through enzyme catalyzed mechanisms (66). The addition of 5 mM thiourea and 1 mM glutathione significantly ($P$-values compared to the control of 0.018 and 0.032) increased the mean survival of *E. coli* MRE162 when levitated for 20 min in LB broth droplets in 50% RH air (Fig. 4C). The addition of water alone did not influence airborne viability (Fig. S4). The consistent improvement in survival upon antioxidant addition, coupled with the diminished airborne survival of K12 $\Delta$SodA, indicate a role of ROS in the airborne loss of viability of *E. coli* within the 20-minute measurement time.

Many forms of stress have been demonstrated to result in the formation of ROS within bacterial cells (46, 67–69), but the measurements with antioxidants indicate that the ROS causing damage to the bacteria were being formed extracellularly. While uptake of small molecules such as thiourea and glutathione by *E. coli* is possible, and is a well-characterized process for glutathione (70), it seemed doubtful that the 20 min of levitation would allow sufficient time for this uptake to impact the viability. To further verify the extracellular nature of the ROS formation, the measurements with 1 mM glutathione were repeated. However, the suspension of bacteria with the glutathione was placed in a 37°C incubator for varying lengths of time prior to levitation to allow the antioxidant to enter the bacteria prior to levitation (Fig. 4D). This preincubation did not result in an improvement to the airborne viability, perhaps indicating that the impact of glutathione was not contingent of the glutathione needing to enter the bacterial cell prior to levitation. Taken together, the measurements with antioxidants suggest a role of ROS in the airborne loss of viability of *E. coli* and that the ROS causing the loss of viability could be formed in the droplet rather than within the bacteria.

## DISCUSSION

The measurements presented provide insights into the mechanisms driving airborne loss of viability in *E. coli*. Osmotic stress, initially an obvious candidate for driving the loss of viability, appears unlikely to drive a loss of viability at the rates observed in aerosols, especially at higher RHs. While in previous measurements with SARS-CoV-2, efflorescence of the droplet appeared to correspond to a large loss of virus in the droplet (33), this was not the case for *E. coli*, with no significant differences in survival in aqueous versus effloresced droplets being observable. Instead, it appears that the presence of oxygen around the droplet is one of the contributing factors, allowing the formation of ROS at the air-liquid interface, which gradually damage and kill the bacteria. The rate at which this happens is determined both by the temperature and the surface area-to-volume ratio of the droplet, with smaller droplets facilitating greater exposure of the bacteria to potential oxidative damage. It can be speculated that while the low water activity of the droplet alone does not have an effect, it could play a role in increasing the rate of oxidative damage by preventing the bacteria from producing the enzymes needed to protect itself against oxidants and by reducing the activity of those enzymes already present within the bacteria. Previous studies have demonstrated increased ROS concentrations in plants experiencing dehydration (71–74), and it is thought that this also occurs in bacterial cells (45).

There are still several unanswered questions surrounding the loss of viability in airborne *E. coli*. While it can be speculated that unique air-liquid interface chemistry will facilitate the formation of ROS from gaseous oxygen at the surface of the droplet, the precise nature of this mechanism remains a mystery. There are also potential mechanisms of bacterial airborne viability loss that were not explored in this study. As airborne droplets evaporate and shrink, it is likely that their pH will change. While relatively stable from pH 4 to 10, *E. coli* will likely experience a loss of viability in airborne droplets that reach extremes of acidity or alkalinity (75, 76). However, the pH change an evaporating droplet of LB broth will undergo upon aerosolization is not yet known, and difficult to measure making the influence of pH on viability loss in the system studies here difficult to investigate. Exhaled aerosol becomes alkaline upon exposure to ambient $CO_2$ concentrations due to the bicarbonate content of respiratory secretions (33, 77, 78), which is a process not replicated in LB broth, but could be explored in future studies of bacteria more likely to be present in exhaled aerosol, such as *Streptococcus* spp. It is also unknown if an enrichment of bacterial cells at the surface of the droplet could play a role in their loss of viability. ROS have very short half-lives (79), meaning that if they are formed at the air-liquid interface, the risk to the bacteria is likely greater at the droplet surface. The role of the surface area-to-volume ratio of the droplet could therefore be 2-fold, both presenting a greater proportion of the droplet to the ROS forming conditions and increasing the likelihood of the bacteria being exposed to the ROS. The degree of contact between bacteria and the droplet surface will also be affected both by the viscosity of the droplet and the motility of the bacteria.

This systematic approach provides clear insights into the possible mechanisms of airborne damage to bacteria, a phenomenon that has been observed for over 70 years (40) but does not yet have an agreed upon mechanistic explanation. While *E. coli* in droplets of LB broth is limited in terms of its relevance to real world biological phenomena, the focus on a well understood and highly controlled system has made possible deeper insights into relationships between the physicochemical properties of the droplet and the biological response of the bacteria. The results of this study provide a starting point for the exploration of more complex biologically relevant systems and will allow for future measurements with different bacterial species in different droplet compositions, to be interpreted in the context of specific processes, such as oxidative and osmotic stress. Such measurements will improve our understanding of what remains among the most mysterious processes in microbiology, and will not only give us useful insights into airborne disease transmission, but will also deepen our understanding of bacterial ecology as a whole.

## MATERIALS AND METHODS

**Strains and growth conditions.** *E. coli* MRE162 and K12 were used for these experiments. K12 isogenic mutants were obtained from the Yale University Keio collection. Bacteria were typically grown at 37°C for 18 h to 24 h in LB broth (NaCl 10 g L$^{-1}$, tryptone 10 g L$^{-1}$, yeast extract 5 g L$^{-1}$, dissolved in distilled water). Broth cultures were incubated for 20 h to 24 h, such that bacteria would reach early to midstationary phase of growth prior to measurement.

**Measuring viability in aerosols.** Bacterial culture was diluted to OD600 0.5 in LB broth (or PBS) prior to the experiment. The diluted culture was then loaded into the droplet dispenser, which was aligned with the electrodynamic balance (EDB) in the center of the CELEBS chamber. Several droplets (typically four per measurement) with a radius of 24 to 28 $\mu$m each, were injected through an induction electrode and into the EDB. The charge applied to the droplets by the induction electrode allowed for the droplets to be trapped in an oscillating electric field created by the ring electrodes of the EDB. The LabVIEW software used to control CELEBS changes the frequency of the field over time, which allowed for droplets to be kept in the field while they evaporated, and their mass to charge ratio changed. This principle has been previously described (3, 4).

The atmosphere around the EDB was maintained at the desired relative humidity through an air inlet angled toward the EDB, the humidity of which can be adjusted by altering the ratio of dry to wet air. The RH of the airflow was measured by a Vaisala HMT330 RH probe prior to entering the levitation chamber, which was used to ensure the airflow entering the chamber was within 1% of the desired RH. The air supply from the compressor could be replaced with a supply of pure nitrogen from a pressurized cylinder to create a hypoxic environment around the droplets. A plate containing LB agar was kept in a compartment beneath the EDB, and a small amount of liquid LB broth was applied to the surface of the plate. After the droplets had been suspended for the desired length of time, the compartment was opened, causing the droplets to deposit into the broth on the surface of the plate. The plate was then incubated at room temperature for approximately 5 min before the broth was spread evenly across the surface and placed into an incubator overnight at 37°C.

**Calculating bacteria per droplet and percentage survival.** The number of bacterial colonies on the agar plate for each levitation was counted the day after the experiment to give the CFU for each levitation. A camera was positioned above the EDB, allowing for the number of droplets trapped for each measurement to be counted and recorded. The CFU per droplet for each levitation was calculated by dividing the number of CFU on the plate by the number of droplets deposited onto the plate. For each experiment, several 5-s levitations were performed to measure the initial CFU per droplet allowing for the results of each experiment to be expressed in terms of percentage survival, making comparison of results between experiments easier.

**Measuring bulk phase changes in viability.** Bacterial cultures were grown overnight as standard. The cultures were then centrifuged for 5 min with a relative centrifugal force of 8,200 × *g*. The resulting pellets were then resuspended in the test solutions. The viability of bacteria in the solution over time was assessed by removing 100 $\mu$L from the solution at each time point and diluting it into LB broth at a factor of 10$^6$. The number of bacteria within this solution was then assessed by CFU counting. The CFU counts were normalized to a measurement taken at the start of the experiment to allow for values to be expressed in terms of percentage survival.

**Statistical analysis.** For each comparison, an F-test was used to determine if the variance of the two data sets was equal. Depending on the results of the F-test, *P* values were calculated using a Student's *t* test either accounting for equal or unequal variance. In the case of multiple comparisons to a control or comparisons between time course data sets, an ANOVA is first carried out to test for significance, followed by multiple t-tests with alpha values adjusted using the Bonferroni-Holm correction for multiple comparisons.

**Data availability.** The txt file data have been deposited in data.bris, the University of Bristol Research Data repository: https://doi.org/10.5523/bris.1u96yv19o3ojh2ow3zpttwp3e4.

## SUPPLEMENTAL MATERIAL

Supplemental material is available online only.
**SUPPLEMENTAL FILE 1**, PDF file, 0.2 MB.

## ACKNOWLEDGMENTS

This work was supported by funding from EPSRC and DSTL. We thank Thomas Hilditch for his suggestion to alter initial solute concentration, and Alexander Hughes-Games for his advice regarding potential airborne death mechanisms.

We declare no conflicts of interest.

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
