## [Reviewer comments · Microbiology Spectrum]

Microbiology Spectrum

Oxidative stress contributes to bacterial airborne loss of viability

Henry Oswin, Allen Haddrell, Cordelia Hughes, Mara Otero Fernandez, Richard Thomas, and Jonathan Reid

Corresponding Author(s): Henry Oswin, University of Bristol

Review Timeline:

Submission Date:	August 24, 2022
Editorial Decision:	November 9, 2022
Revision Received:	January 4, 2023
Accepted:	January 18, 2023

Editor: Jannell Bazurto

Reviewer(s): The reviewers have opted to remain anonymous.

Transaction Report:

DOI: <https://doi.org/10.1128/spectrum.03347-22>

November 9, 2022

Mx. Henry Oswin
University of Bristol
Chemistry
Bristol
United Kingdom

Re: Spectrum03347-22 (Oxidative stress contributes to bacterial airborne loss of viability)

Dear Mx. Henry Oswin:

Thank you for submitting your manuscript to Microbiology Spectrum. I have received two reviews of your manuscript. Both agreed that this is an interesting and well-executed study, applying newer approaches to glean new insights into the role of oxidative stress in airborne loss of viability. Though well-received the reviewers did have some concerns, which should be addressed.

Link Not Available

Sincerely,

Jannell Bazurto

Journals Department
Reviewer comments:

Reviewer #1 (Comments for the Author):

The present study examines potential mechanisms responsible for losses of viability observed in *E. coli* in aerosols. The experiments are well thought out and carried out using appropriate methodologies, and the manuscript is generally well-written and referenced. Several novel results are presented, especially relating to the role of particle size/surface area to volume ratio, which provide insights into the processes involved. While several of the observed results are similar to findings reported in previous studies, the methods utilized are more refined than those utilized in some of these older studies, including the

electrodynamic balance for capturing and holding particles, and the use of genetic mutants to evaluate the role of oxidative stress, making this a valuable confirmation and extension of those previous studies. I feel that the work encompassed in the manuscript is valuable and contributes novel information to study of the survival of microorganisms in aerosol particles. However, I do have some concerns related to some of the experiments and statistical analysis/ data presentation that I feel need to be addressed prior to publication. I have included more detailed comments in the attached file.

Reviewer #2 (Comments for the Author):

A very good paper. I have only one minor observation which i would like clarified.

You used mutants to investigate the protective effect of trehalose in a 10 minute period. Surely the gene would not be upregulated due to low water activity during this period of time so you would not expect any effect. Could you not have added trehalose to the spray suspension?

Staff Comments:

Preparing Revision Guidelines

Please return the manuscript within 60 days; if you cannot complete the modification within this time period, please contact me. If you do not wish to modify the manuscript and prefer to submit it to another journal, please notify me of your decision immediately so that the manuscript may be formally withdrawn from consideration by Microbiology Spectrum.

Oxidative stress contributes to bacterial airborne loss of viability

Henry P. Oswin^{1*}, Allen E. Haddrell¹, Cordelia Hughes¹, Mara Otero-Fernandez¹, Richard J. Thomas², Jonathan P. Reid^{1*}

The present study examines potential mechanisms responsible for losses of viability observed in *E. coli* in aerosols. The experiments are well thought out and carried out using appropriate methodologies, and the manuscript is generally well-written and referenced. Several novel results are presented, especially relating to the role of particle size/surface area to volume ratio, which provide insights into the processes involved. While several of the observed results are similar to findings reported in previous studies, the methods utilized are more refined than those utilized in some of these older studies, including the electrodynamic balance for capturing and holding particles, and the use of genetic mutants to evaluate the role of oxidative stress, making this a valuable confirmation and extension of those previous studies. I feel that the work encompassed in the manuscript is valuable and contributes novel information to study of the survival of microorganisms in aerosol particles. However, I do have some concerns related to some of the experiments and statistical analysis/ data presentation that I feel need to be addressed prior to publication. I have included more detailed comments below.

SPECIFIC COMMENTS

There are no page number or line numbers in the provided documents, so my comments are referenced by each section and sub-heading.

Results – Characterisation of the response of *E. coli* to the airborne environment:

MAJOR: It is stated that both strains lose culturability after 20 minutes, decreasing to $30\pm 37\%$ for K12 and $57\pm 17\%$ for MRE 162, and that the K12 is “notably less stable.” No statistical analysis is presented for this statement – What was the p-value? What statistical comparison was done? Are these values the mean and standard deviation? I noticed later that there was text on data analysis once I discovered the methods section in the supplementary material, but I would recommend something be included here as well as it will help the reader interpret the results. This should be done throughout the results, as there are many instances where it is stated or inferred that there are differences (some additional instances are noted later), but p-values and the comparisons made/statistical tests utilized are not stated.

MINOR: For the data presented in Figure 1, panels A through C present the data as mean \pm standard error, while panel D presents the data as mean \pm standard deviation. Was there a particular reason the standard error is presented? I would recommend using standard deviation throughout so the reader can assess the variability associated with the measurements, as opposed to the standard error, which is assessing the precision of the sample mean relative to the population mean.

MINOR: For the data presented in Figure 1, panels A and B, the authors are making measurements at multiple time points to assess losses in viability. However, the comparisons presented only compare the first and last values, which ignores all of intervening data points. Have the author's considered fitting a model to the timecourse data to estimate the rate at which viability is being lost? I don't imagine this would change any conclusions derived from the data, but it would incorporate all of the data generated into the analysis.

MINOR: I would recommend providing a reference for the statement: "as it is well established that *E. coli* do not lose viability in bulk solutions of LB broth at these temperatures."

MINOR: It is stated at the end of the section that the values here are different than a previous publication from the same group due to improvements in the methodology since the previous paper. SI Section 1 describes these changes. The reader is asked to compare Figure 1a from the present manuscript to Figure 3a from the previous paper. While the graphs do appear different, it would be useful to include the mean values and p-value from a statistical comparison of the data from the two studies, which presumably are available to the authors since both studies were performed in the same laboratory.

Results – Airborne droplet efflorescence does not appear to impact the viability of *E. coli*:

Very interesting finding that the phase change does not seem to impact survival.

MINOR: Please include a p-value for the comparisons presented in Figure 2B.

Results – Dehydration is unlikely to contribute to the airborne loss of viability of *E. coli*:

MAJOR: It is stated that "trehalose is synthesised by *E. coli* in response to low water activity conditions." Is it known how fast this synthesis occurs? The data presented are viability following 2- or 10-minute levitations. Do the authors think this timeframe is sufficient for the WT bacteria to synthesize sufficient trehalose for protection? If not, then the results with the K12 Δ OtsA are not unexpected.

Additionally, it is stated that "it can be hypothesised that deletion of the OtsA gene in *E. coli*, shown to significantly lower intracellular concentrations of trehalose, would diminish airborne survival if osmotic stress is a major contributor to airborne loss of viability in *E. coli*." When were the lower concentrations of trehalose measured in the cited studies – are these basal levels in culture or following exposure to low water activity? Is it known what the basal levels of trehalose are in K12 WT vs Δ OtsA in culture?

MAJOR: It is noted that the viability at 1-hr was decreased in bulk solution with a water activity of 0.76 relative to 0.95. What statistical comparison was done here? In the Materials and Methods, it is stated that a t-test was used for comparisons. However, this would require multiple t-tests be performed as there were presumably four different comparisons done (comparison of the response for each water activity at each time point). Given this experimental design, ANOVA is more appropriate, as it avoids increase in the Type I error probability encountered with multiple t-tests.

Results – Loss of airborne viability correlated with surface area-to-volume ratio.

The data presented in Figure 3C and 3D are quite compelling and demonstrate the importance of surface area to volume ratio. Additionally, I was glad to see the following included – “As a minor consideration, the different starting concentrations will also lead to differences in the evaporation kinetics although these differences are small and occur over a time period during which no loss of viability is observed (see Fig. S1 for simulations of the evaporation kinetics of these different starting solution droplets).”

MINOR: However, in Figure S1, please add a reference or more detail on the data utilized to inform the modeling presented.

Results – *E. coli* airborne loss of viability is reduced in a hypoxic environment.

MINOR: Please include p-values, either in the text or figure, for the statement: “In all cases, replacing the gas flow with nitrogen increased the average airborne viability of *E. coli* MRE162.”

MINOR: In addition of the references already included, please consider including and discussing the following reference from the food industry, as it is much more recent than those already cited, and reports many of the same effects observed in the present study (i.e. a pure nitrogen atmosphere and addition of scavengers diminish losses) related to the potential role of oxidative stress in bacterial damage:

Ghandi, Amir, et al. "Effect of shear rate and oxygen stresses on the survival of *Lactococcus lactis* during the atomization and drying stages of spray drying: a laboratory and pilot scale study." *Journal of food engineering* 113.2 (2012): 194-200.

MINOR: It is stated that “However, if this were the case it seems unlikely that a loss of viability would take place in PBS droplets (as seen in Figure 2) where bacteria would be starved of the nutrients needed for aerobic respiration.” How long before levitation were bacteria re-suspended in PBS? Is this time sufficient to “starve” them of nutrients?

MINOR: Should the following statement be referencing Figure 3C, and not 3D? “If the loss of viability was a combination of the high concentration of solutes, bacteria, and the presence of oxygen, a more significant loss of viability would likely have been observed in the bulk solution of Figure 3D”

Results – Reactive oxygen species formation drives airborne loss of viability in *E. coli*.

“it is also possible that reducing the capacity of the bacteria to process reactive oxygen species (ROS) merely introduced oxidative stress as an additional mechanism of viability loss.” This is a good point to include here and sets up the next set of experiments with the free radical scavengers nicely.

MINOR: For figure S3, it would be worth including additional detail on the various mutants and why they were included. "It was, however, noteworthy that of all the isogenic mutants of K12 studied, it was only the Δ SodA mutant that demonstrated reduced airborne viability (Fig. S3)." This is an interesting point. Is it known where the SOD resides within the bacterium relative to the proteins altered in the other mutants? Several references suggest SOD resides in the periplasmic space, which would possibly provide additional evidence for the hypothesis that the cause of damage due to oxidative processes originates extracellularly, especially if the alterations in the other mutants were all intracellular.

MAJOR: The text states that "The addition of all three of these antioxidants increased the mean survival of *E. coli* MRE162 when levitated for 20 minutes in LB broth droplets in 50% RH air (Fig. 4C).", whereas in figure 4C, it is noted that the increase for SOD is not statistically significant. Please revise this text to make this consistent. I would recommend just stating that there were significant increases for 3 of the 4 scavengers added. Simply saying the mean increased is not appropriate statistically. Additionally, what statistical test was done here – multiple t-tests as suggested by the methods? As noted earlier, ANOVA is more appropriate given the number of comparisons.

MAJOR: The claim of dose-dependence for the glutathione effect is questionable. "There was suggestion of a dose dependent effect for glutathione, with 1mM glutathione increasing the viability to $70\pm 4\%$ and 5mM increasing the viability to $79\pm 23\%$." Were the two values significantly different from one another? Perhaps a more appropriate analysis would be to perform regression analysis with the data points from the control (0 mM), 1 mM, and 5mM groups and see if there is a significant non-zero slope to the line? If not, you cannot make the claim of dose-dependence.

MINOR: Why did the authors choose to examine the effect of the various antioxidants shown in Figure 4C at 50% RH? Wouldn't using 30% would have provided a better opportunity to quantify an increase in survival since the control group at the lower RH had a greater loss of viability?

MAJOR: I do not agree with the logic presented in paragraph containing: "Furthermore, whilst it was not as significant compared to the other antioxidants ($p=0.28$), the average viability did increase upon addition of superoxide dismutase, which would be unable to enter the bacterial cell." Simply comparing the mean values to say viability was increased is not appropriate statistically. While the viability was higher, it was not significant (as the p -value of 0.28 indicates), so you cannot claim it increased. Thus, the hypothesis that an increase in viability with addition of SOD suggests that the origin of the oxidative stress is extracellular does not follow. In fact, one could argue the opposite – i.e. all of the antioxidants except SOD has a modest effect to increase viability. Thus, the origin of the oxidative insult is likely intracellular since SOD is not able to enter the cells.

MAJOR: What is known about the uptake rate of glutathione by *E. coli*? Are there data in the literature to support the durations chosen? If so, some additional text should be added for support, especially given the lack of effect (although the variability are quite high, making it difficult to detect a significant difference). Additionally, are the data in figure 4D significantly different from the control data from Figure 4C when compared by ANOVA?

SI Section 1:**Supplementary Information:**

MAJOR: Materials and Methods: According to the journal guidelines for authors, the materials and methods should be included in the main text, and do not count against the 5000 word count limit. However, in the manuscript's present form, they are included as Supplementary Information. In my first read through the paper, I was left wondering where the methods were as I had not yet looked at the Supplementary Information, and their location was not referenced in the main text. I would strongly recommend moving these to the main text.

MINOR: Do the authors have any data to demonstrate that 5 minutes is the optimal time for disaggregation of particles before spread plating?

Subheadings are bold and underlined

Reviewer comments are written in standard font.

Author responses are written in italics

"Text edits from the paper are written in italics and highlighted"

Response to Reviewer 1

Author comment: The authors thank the reviewer for their very thorough and useful review. We feel that their comments will strengthen the paper and help us to better present our findings.

Many of the reviewer's comments were in reference to the statistical analysis used to demonstrate significant effects of different parameters. Particularly, in the case of timecourse data comparison. For such datasets, we have now taken the approach of initially using an ANOVA to test for any significance. If the ANOVA finds a significant difference, this is followed by the use of multiple t-test comparisons, with a t-test being used to compare each timepoint. To reduce the risk of type 1 error, a Bonferroni-Holm correction is used to adjust the alpha values used to determine significance in these cases. The statistical analysis section of the methods has been edited to say the following:

"For each comparison, an F-test was used to determine if the variance of the two datasets was equal. Depending on the results of the F-test, p values were calculated using a student's t-test either accounting for equal or unequal variance. In the case of multiple comparisons to a control or comparisons between time course datasets, an ANOVA is first carried out to test for significance, followed by multiple t-tests with alpha values adjusted using the Bonferroni-Holm correction for multiple comparisons."

Results – Characterisation of the response of E. coli to the airborne environment:

MAJOR: It is stated that both strains lose culturability after 20 minutes, decreasing to 30±37% for K12 and 57±17% for MRE 162, and that the K12 is "notably less stable." No statistical analysis is presented for this statement – What was the p-value? What statistical comparison was done? Are these values the mean and standard deviation? I noticed later that there was text on data analysis once I discovered the methods section in the supplementary material, but I would recommend something be included here as well as it will help the reader interpret the results. This should be done throughout the results, as there are many instances where it is stated or inferred that there are differences (some additional instances are noted later), but p-values and the comparisons made/statistical tests utilized are not stated.

Author response: In addition to the change described above that presents a general statement of the statistical tests performed, we now include the following statement on this specific comparison:

"but K12 is significantly less stable (p=0.009 at 120 seconds, p=0.0002 at 300 seconds, and p=0.002 at 600 seconds)"

MINOR: For the data presented in Figure 1, panels A through C present the data as mean \pm standard error, while panel D presents the data as mean \pm standard deviation. Was there a particular reason the standard error is presented? I would recommend using standard deviation throughout so the reader can assess the variability associated with the measurements, as opposed to the standard error, which is assessing the precision of the sample mean relative to the population mean.

Author response: All error bars throughout the paper have now been altered to show standard deviation rather than standard error.

MINOR: For the data presented in Figure 1, panels A and B, the authors are making measurements at multiple time points to assess losses in viability. However, the comparisons presented only compare the first and last values, which ignores all of intervening data points. Have the author's considered fitting a model to the timecourse data to estimate the rate at which viability is being lost? I don't imagine this would change any conclusions derived from the data, but it would incorporate all of the data generated into the analysis.

Author response: Whilst this would be interesting, for the purposes of this paper, we do not feel such detailed analysis are necessary. The data presented in Figure 1 serves mainly to provide context for the later investigation of the mechanisms driving the viability loss. Assumptions would need to be made about the decay mechanism (e.g. leading to a single exponential decay with single time-constant) and we prefer not to make such assumptions at this stage.

MINOR: I would recommend providing a reference for the statement: "as it is well established that *E. coli* do not lose viability in bulk solutions of LB broth at these temperatures."

Author response: Reference added to Escherichia coli Physiology in Luria-Bertani Broth by Sezonov et al.

MINOR: It is stated at the end of the section that the values here are different than a previous publication from the same group due to improvements in the methodology since the previous paper. SI Section 1 describes these changes. The reader is asked to compare Figure 1a from the present manuscript to Figure 3a from the previous paper. While the graphs do appear different, it would be useful to include the mean values and p-value from a statistical comparison of the data from the two studies, which presumably are available to the authors since both studies were performed in the same laboratory.

Author response: We have added a figure to the supplementary information clearly showing the improvement to measurements granted by the changes to the protocol. The following text has been added to the SI to refer to the figure:

"A comparison of survival measurements using both the old and new protocol to measure the decay of *E. coli* MRE162 at 70% RH is shown in Figure S1."

Results – Airborne droplet efflorescence does not appear to impact the viability of E. coli:

Very interesting finding that the phase change does not seem to impact survival.

MINOR: Please include a p-value for the comparisons presented in Figure 2B.

Author response: A p-value calculated by ANOVA has now been included in the figure.

Results – Dehydration is unlikely to contribute to the airborne loss of viability of E. coli:

MAJOR: It is stated that “trehalose is synthesised by E. coli in response to low water activity conditions.” Is it known how fast this synthesis occurs? The data presented are viability following 2- or 10-minute levitations. Do the authors think this timeframe is sufficient for the WT bacteria to synthesize sufficient trehalose for protection? If not, then the results with the K12 Δ OtsA are not unexpected. Additionally, it is stated that “it can be hypothesised that deletion of the OtsA gene in E. coli, shown to significantly lower intracellular concentrations of trehalose, would diminish airborne survival if osmotic stress is a major contributor to airborne loss of viability in E. coli.” When were the lower concentrations of trehalose measured in the cited studies – are these basal levels in culture or following exposure to low water activity? Is it known what the basal levels of trehalose are in K12 WT vs Δ OtsA in culture?

Author response: Whilst OtsA expression is upregulated in response to various stresses, it is always expressed to some degree and E. coli will contain intracellular trehalose prior to levitation. An additional reference to Kandror 2002 has been added that more clearly shows intracellular trehalose concentrations and the influence of OtsA deletion. Additionally, the opening line of this paragraph has been edited to the following:

“The disaccharide trehalose serves as an osmoprotectant for E. coli and its production rate is increased in response to low water activity conditions”

The original line may have misled readers into thinking trehalose is absent until the bacteria are exposed to low water activity, which is not the case.

We have not yet found any conclusive evidence suggesting that there could be an increase in intracellular trehalose during 10-minutes of levitation for the WT K12, but it is clear from the literature that the levitated WT bacteria will contain more trehalose than the mutant.

MAJOR: It is noted that the viability at 1-hr was decreased in bulk solution with a water activity of 0.76 relative to 0.95. What statistical comparison was done here? In the Materials and Methods, it is stated that a t-test was used for comparisons. However, this would require multiple t-tests be performed as there were presumably four different comparisons done (comparison of the response for each water activity at each time point). Given this experimental design, ANOVA is more appropriate, as it avoids increase in the Type I error probability encountered with multiple t-tests.

Author response: It was a mistake to suggest that there was a difference here. Multiple t-tests were performed but in the absence of a multiple comparison correction. When the

alpha value was adjusted using the Bonferroni-Holm method, the difference was no longer significant. The text has been edited to the following:

"A small reduction in the mean CFU count was observable in both solutions over three hours, both falling to approximately 80% of the initial count, but there was no significant difference in culturability of *E. coli* between the two suspensions."

The change to this result now shows that decreased water activity does not result in an increased loss of viability in bulk solution, serving to strengthen the conclusion that dehydration does not contribute to airborne loss of viability.

Results – Loss of airborne viability correlated with surface area-to-volume ratio.

The data presented in Figure 3C and 3D are quite compelling and demonstrate the importance of surface area to volume ratio. Additionally, I was glad to see the following included – "As a minor consideration, the different starting concentrations will also lead to differences in the evaporation kinetics although these differences are small and occur over a time period during which no loss of viability is observed (see Fig. S1 for simulations of the evaporation kinetics of these different starting solution droplets)."

MINOR: However, in Figure S1, please add a reference or more detail on the data utilized to inform the modeling presented.

We have added a citation to the paper describing these data along with the following text:

"using previously collected physicochemical data²¹"

Results – *E. coli* airborne loss of viability is reduced in a hypoxic environment.

MINOR: Please include p-values, either in the text or figure, for the statement: "In all cases, replacing the gas flow with nitrogen increased the average airborne viability of *E. coli* MRE162."

Author response: The following was added:

"but only at 30% RH was a significant difference observable ($p < 0.01$)"

MINOR: In addition of the references already included, please consider including and discussing the following reference from the food industry, as it is much more recent than those already cited, and reports many of the same effects observed in the present study (i.e. a pure nitrogen atmosphere and addition of scavengers diminish losses) related to the potential role of oxidative stress in bacterial damage: Ghandi, Amir, et al. "Effect of shear rate and oxygen stresses on the survival of *Lactococcus lactis* during the atomization and drying stages of spray drying: a laboratory and pilot scale study." *Journal of food engineering* 113.2 (2012): 194-200.

Author response: We have added the citation as well as the following text:

"as well as a more recent study of the effects of spray drying on *Lactococcus lactis*"

MINOR: It is stated that “However, if this were the case it seems unlikely that a loss of viability would take place in PBS droplets (as seen in Figure 2) where bacteria would be starved of the nutrients needed for aerobic respiration.” How long before levitation were bacteria re-suspended in PBS? Is this time sufficient to “starve” them of nutrients?

Author response: This statement was not needed to make the arguments in the paragraph, and it was difficult to find good references supporting it, so we have removed it from the text.

MINOR: Should the following statement be referencing Figure 3C, and not 3D? “If the loss of viability was a combination of the high concentration of solutes, bacteria, and the presence of oxygen, a more significant loss of viability would likely have been observed in the bulk solution of Figure 3D”

Author response: The reviewer is correct. We have now edited the figure reference to state 3C.

Results – Reactive oxygen species formation drives airborne loss of viability in E. coli.

“it is also possible that reducing the capacity of the bacteria to process reactive oxygen species (ROS) merely introduced oxidative stress as an additional mechanism of viability loss.” This is a good point to include here and sets up the next set of experiments with the free radical scavengers nicely.

MINOR: For figure S3, it would be worth including additional detail on the various mutants and why they were included. “It was, however, noteworthy that of all the isogenic mutants of K12 studied, it was only the Δ SodA mutant that demonstrated reduced airborne viability (Fig. S3).” This is an interesting point. Is it known where the SOD resides within the bacterium relative to the proteins altered in the other mutants? Several references suggest SOD resides in the periplasmic space, which would possibly provide additional evidence for the hypothesis that the cause of damage due to oxidative processes originates extracellularly, especially if the alterations in the other mutants were all intracellular.

Author response: This is an interesting point. However, whilst SodA is present within the periplasm, it is also abundant within the cytosol meaning its knockout could also be impacting intracellular ROS formation. By contrast, SodC does appear to be present at higher concentrations within the periplasm and so investigating the airborne stability of SodC knockouts could be a future experiment we carry out to further investigate the source of the ROS.

MAJOR: The text states that “The addition of all three of these antioxidants increased the mean survival of E. coli MRE162 when levitated for 20 minutes in LB broth droplets in 50% RH air (Fig. 4C).”, whereas in figure 4C, it is noted that the increase for SOD is not statistically significant. Please revise this text to make this consistent. I would recommend just stating that there were significant increases for 3 of the 4 scavengers added. Simply saying the mean increased is not appropriate statistically. Additionally, what statistical test

was done here – multiple t-tests as suggested by the methods? As noted earlier, ANOVA is more appropriate given the number of comparisons.

Author response: Statistical analysis has been adjusted based on the reviewer's advice and the text has been edited to say the following:

"The addition of 5mM thiourea and 1mM glutathione significantly (p-values compared to the control of 0.018 and 0.032) increased the mean survival of E. coli MRE162"

MAJOR: The claim of dose-dependence for the glutathione effect is questionable. "There was suggestion of a dose dependent effect for glutathione, with 1mM glutathione increasing the viability to 70±4% and 5mM increasing the viability to 79±23%." Were the two values significantly different from one another? Perhaps a more appropriate analysis would be to perform regression analysis with the data points from the control (0 mM), 1 mM, and 5mM groups and see if there is a significant non-zero slope to the line? If not, you cannot make the claim of dose-dependence.

Author response: The claim of a dose dependence has been removed.

MINOR: Why did the authors choose to examine the effect of the various antioxidants shown in Figure 4C at 50% RH? Wouldn't using 30% would have provided a better opportunity to quantify an increase in survival since the control group at the lower RH had a greater loss of viability?

Author response: 50% RH is used as a starting point for all our measurements of airborne stability. 50% is typical of indoor RH, which makes it readily accessible, even in more rudimentary experimental setups, and also makes it relevant to many typical scenarios in which bioaerosols may be generated, particularly those relevant to disease transmission. It is hoped that in the future it may be possible to expand this investigation over a broader range of conditions.

MAJOR: I do not agree with the logic presented in paragraph containing: "Furthermore, whilst it was not as significant compared to the other antioxidants (p=0.28), the average viability did increase upon addition of superoxide dismutase, which would be unable to enter the bacterial cell." Simply comparing the mean values to say viability was increased is not appropriate statistically. While the viability was higher, it was not significant (as the p-value of 0.28 indicates), so you cannot claim it increased. Thus, the hypothesis that an increase in viability with addition of SOD suggests that the origin of the oxidative stress is extracellular does not follow. In fact, one could argue the opposite – i.e. all of the antioxidants except SOD has a modest effect to increase viability. Thus, the origin of the oxidative insult is likely intracellular since SOD is not able to enter the cells.

Author response: We agree with the reviewer here. The sentence referring to the increase in survival upon addition of SOD has now been removed.

MAJOR: What is known about the uptake rate of glutathione by E. coli? Are there data in the literature to support the durations chosen? If so, some additional text should be added for support, especially given the lack of effect (although the variability are quite high, making it

difficult to detect a significant difference). Additionally, are the data in figure 4D significantly different from the control data from Figure 4C when compared by ANOVA?

Author response: Upon further inspection of this data, we agree that it is not accurate to say that the survival was reduced after preincubation and the text has been edited to say the following:

“This pre-incubation did not result in an improvement to the airborne viability, perhaps indicating that the impact of glutathione was not contingent of the glutathione needing to enter the bacterial cell prior to levitation.”

However, it is still accurate to say that there was no increase in survival upon preincubation, which does support the conclusion that the antioxidants do not need to enter the bacteria in order to protect them from airborne viability loss.

*The question of how long a preincubation is enough time to allow sufficient glutathione uptake to occur to protect the bacteria is difficult to answer. Whilst data exists regarding glutathione uptake (and is cited) confirming that some glutathione uptake will occur on this timescale, it is not possible to conclude from those measurements how much glutathione uptake would need to occur to protect *E. coli* from oxidative stress. We have attempted to soften the language throughout this paragraph to attempt to account for this uncertainty, as indicated in the quoted text above and also in the following statement:*

“the ROS causing the loss of viability could be formed in the droplet”

SI Section 1: Supplementary Information:

MAJOR: Materials and Methods: According to the journal guidelines for authors, the materials and methods should be included in the main text, and do not count against the 5000 word count limit. However, in the manuscript's present form, they are included as Supplementary Information. In my first read through the paper, I was left wondering where the methods were as I had not yet looked at the Supplementary Information, and their location was not referenced in the main text. I would strongly recommend moving these to the main text.

Author response: This was an oversight on the part of the authors. The Materials and Methods have now been moved back into the main text.

MINOR: Do the authors have any data to demonstrate that 5 minutes is the optimal time for disaggregation of particles before spread plating?

Author response: Confidence in this protocol came from comparing the expected CFU per droplet (as calculated using the CFU per ml of the suspension loaded into the dispenser and the expected volume of the droplets) to the actual CFU per droplet from the measurements. As the measured CFU per droplet was as expected using a 5-minute wait before spreading the broth across the plate, it was assumed that this time was sufficient. Additionally, there were practical considerations in deciding upon a 5-minute wait time. 5-minutes is sufficient time to remove the plate from the instrument and begin the next levitation, allowing for experiments to be carried out in a more streamlined, efficient manner.

Response to Reviewer 2

A very good paper. I have only one minor observation which I would like clarified.

You used mutants to investigate the protective effect of trehalose in a 10-minute period. Surely the gene would not be upregulated due to low water activity during this period of time so you would not expect any effect. Could you not have added trehalose to the spray suspension?

Author response: Whilst OtsA expression is upregulated in response to various stresses, it is always expressed to some degree and E. coli will contain intracellular trehalose prior to levitation. An additional reference to Kandror 2002 has been added that more clearly shows intracellular trehalose concentrations and the influence of OtsA deletion. Additionally, the opening line of this paragraph has been edited to the following:

"The disaccharide trehalose serves as an osmoprotectant for E. coli and its production rate is increased in response to low water activity conditions"

The original line may have misled readers into thinking trehalose is absent until the bacteria are exposed to low water activity, which is not the case.

We have not yet found any conclusive evidence suggesting that there could be an increase in intracellular trehalose during 10-minutes of levitation for the WT K12, but it is clear from the literature that the levitated WT bacteria will contain more trehalose than the mutant.

Adding trehalose (or trehalose 6-phosphate) to the growth medium prior to levitation could provide an interesting means of further investigating the apparent increased survival of the OtsA mutant. Trehalose uptake from the environment is not contingent on OtsA expression, and so the OtsA deletion mutant would contain trehalose if grown in its presence. If prior incubation with trehalose results in the improved airborne survival of the OtsA mutant being reversed, this would indicate that the improved survival is due to the absence of intracellular trehalose rather than a downstream regulatory effect. We may carry out this measurement in the future and thank the reviewer for their suggestion.

January 18, 2023

Mx. Henry Oswin
University of Bristol
Chemistry
Bristol
United Kingdom

Re: Spectrum03347-22R1 (Oxidative stress contributes to bacterial airborne loss of viability)

Dear Mx. Henry Oswin:

Your manuscript has been accepted, and I am forwarding it to the ASM Journals Department for publication. You will be notified when your proofs are ready to be viewed.

Sincerely,

Jannell Bazurto
Editor, Microbiology Spectrum
